# Intracellular Low Iron Exerts Anti-BK Polyomavirus Effect by Inhibiting the Protein Synthesis of Exogenous Genes

Jiajia Sun,[a,b] Yejing Shi,[a] Huichun Shi,[a,b] Yumin Hou,[a] Chunlan Hu,[a] Yigang Zeng,[a] Guoyi Wu,[a] Tongyu Zhu[a,b]

aShanghai Public Health Clinical Center, Fudan University, Shanghai, China
bShanghai Key Laboratory of Organ Transplantation, Shanghai, China

Jiajia Sun, Yejing Shi, and Huichun Shi contributed equally to this article. Author order was determined both alphabetically and in order of increasing seniority.

**ABSTRACT** BK polyomavirus (BKPyV) is a small double-stranded DNA virus and ubiquitous human pathogen that particularly affects immunocompromised individuals. Antiviral therapy for BKPyV is urgently needed. Intracellular irons have an important role in many viral infections, yet its contribution to BKPyV and replication has not been explored. In this study, we explored the interaction between BKPyV infection and intracellular iron and the inhibitory effect of iron depletion on BKPyV infection. By creating a low-intracellular-iron environment, we demonstrated that the iron-chelating-induced iron depletion inhibits BKPyV infection in primary renal tubular epithelial cells (RPTECs) and urinary bladder cancer cells (TCCSUP cells). Iron depletion exerts an inhibitory effect after BKPyV enters the nucleus, which might be due to the inhibition of the protein synthesis of exogenous genes in iron-depleted cells. Further exploration of the target proteins of iron-regulating viral infection could potentially be used to develop new strategies for urgently needed anti-BKPyV therapies.

**IMPORTANCE** BKPyV poses a serious threat to the health of immunocompromised patients, and there are currently no curative drugs. Understanding the relationship between the virus and intracellular environment contributes to the discovery of antiviral targets. We demonstrate here that BKPyV is inhibited in cells with a low-iron environment. We also find that iron-chelating-induced iron depletion inhibits viral and exogenous protein synthesis. Further exploration of the target proteins of iron regulation could have great potential in developing new drugs against BKPyV and other viruses.

**KEYWORDS** BK polyomavirus, BKVAN, iron depletion, protein synthesis

BK polyomavirus (BKPyV) was initially isolated from a recipient with ureteral stenosis after kidney transplantation (1). BKPyV leads to persistent asymptomatic infection in the epithelial cells of the renal tubules and urinary tract (2), which is considered a risk factor for renal transplant dysfunction (3–5). Its reactivation may lead to BKPyV-associated nephropathy (BKVAN), ureteral stenosis, hemorrhagic cystitis, and BKPyV-associated urinary tract tumors in immunosuppressed individuals (5). BKPyV-related diseases are difficult to treat; so far, no BKPyV-specific antiviral drug has been developed. Clinical cohort studies on leflunomide (6), cidofovir (7), or quinolone antibiotics (8) have shown that these drugs are not fully safe or efficient enough to treat BKPyV-related diseases in posttransplant recipients. Intravenous immunoglobulin (IVIG) showed effectiveness against BKPyV, but it was used only as an adjunct to other measures in refractory cases (9, 10). Clinically, therapies that reduce the dose of immunosuppressive agents are generally used to enhance the antiviral immune response to inhibit further development of BKVAN. However, the reduction of immunosuppression increases the risk of acute rejection (11), which further complicates treatment regimens.

Address correspondence to Tongyu Zhu, tyzhu@shphc.org.cn, Guoyi Wu, wuguoyi@163.com, or Yigang Zeng, zengyigang@shphc.org.cn.

BKPyV is a nonenveloped DNA virus belonging to the polyomavirus family. Structurally, BKPyV has an unenveloped, icosahedral capsid composed of capsid proteins VP1, VP2, and VP3, which surround double-stranded DNA molecules that bind to histones in the form of chromatin chains (5). The BKPyV genome consists of two highly conserved coding regions that produce early and late proteins, respectively, separated by a noncoding control region (NCCR), which contains the starting sites for viral DNA replication, as well as promoters that drive the transcription of early and late viral genes (12). Early genes encode large tumor antigens (LTags), small tumor antigens (STags), and truncated Tags, which exert a regulatory role in viral DNA replication and transcription of late coding regions. The late gene encodes structural proteins VP1, VP2, and VP3 and agnoproteins, expressed after initiating genome replication.

BKPyV principally hijacks the host cell DNA replication machinery for its reproduction. Virus reproduction is usually associated with enhanced cell metabolism due to synthesis of many proteins and genome replication. Therefore, understanding the relationship between BKPyV and the host cell environment is important for the development of anti-BKPyV therapeutics.

Iron regulates many cellular physiological, metabolic activities. It is involved in forming mitochondrial electron transport chains and hemoglobin components and participating in the synthesis of iron-dependent enzymes. These cellular functions frequently overlap those hijacked during virus infection. According to previous studies, iron activation is involved in viral RNA reverse transcription (13), mRNA translation (13, 14), virus packaging (13), and activation of nuclear factor kappa-B (NF-$\kappa$B) in HIV-1-infected macrophages (13, 15). In addition, it can promote HCV mRNA translation by enhancing the function of eIF3 and La proteins (16, 17). Altering cellular iron metabolism is also being investigated as a strategy to inhibit these viral infections. *In vitro* studies have demonstrated the efficacy of iron chelators in controlling HIV-1, HCMV, vaccinia virus, herpes simplex virus, and hepatitis B virus replication (13). However, the contribution of the intracellular iron environment to BKPyV infection and replication has not been explored.

In this study, using established BKPyV infection models in primary renal tubular epithelial cells (RPTECs) and urinary bladder cancer (TCCSUP) cells, we explored the interaction between BKPyV infection and intracellular iron and the inhibitory effect of iron depletion on BKPyV infection.

## RESULTS

**BKPyV infection increases intracellular free iron levels.** Viruses hijack cells in order to replicate, and efficient replication needs an iron-replete host. Some viruses can increase available iron levels by regulating iron metabolism in host cells. Here, we examined whether BKPyV infection affects intracellular iron levels. BKPyV infection assay was performed in RPTECs and TCCSUP cells, used as clinicopathologically relevant cell culture infection models. Calcein-AM staining was used to quantitatively compare intracellular quenchable iron pools, which were negatively correlated with intracellular labile iron leveling. In this study, calcein-AM staining was significantly reduced in BKPyV-infected RPTECs and TCCSUP cells relative to that in uninfected cells ($P < 0.05$; Fig. 1A and B), thus indicating that the level of the intracellular labile irons in BKPyV-infected cells was significantly upregulated. These data suggested that BKPyV infection can induce the increase of intracellular free iron ions.

**Iron chelators inhibit BKPyV infection.** To explore the effect of intracellular iron depletion on BKPyV infection, three clinically approved iron chelators, deferasirox (DFX), deferiprone (DFP), and deferoxamine mesylate (DFO), were used to create an intracellular iron-low environment in BKPyV infection assay. BKPyV infection was measured through the VP1 major capsid protein expression, known to correlate with virus production. Iron chelators were added to cells with virus supernatants, permitting the inhibition of all viral life cycle stages. Corresponding data showed that in both RPTEC and TCCSUP cells, the treatment of DFX, DFP, and DFO at 20 $\mu$M concentrations inhibited the protein expression of VP1 to different degrees (Fig. 2A). In addition, the expression of early LTag protein was also inhibited (Fig. 2A), confirming that the effects were not restricted to VP1. Iron chelators also reduced

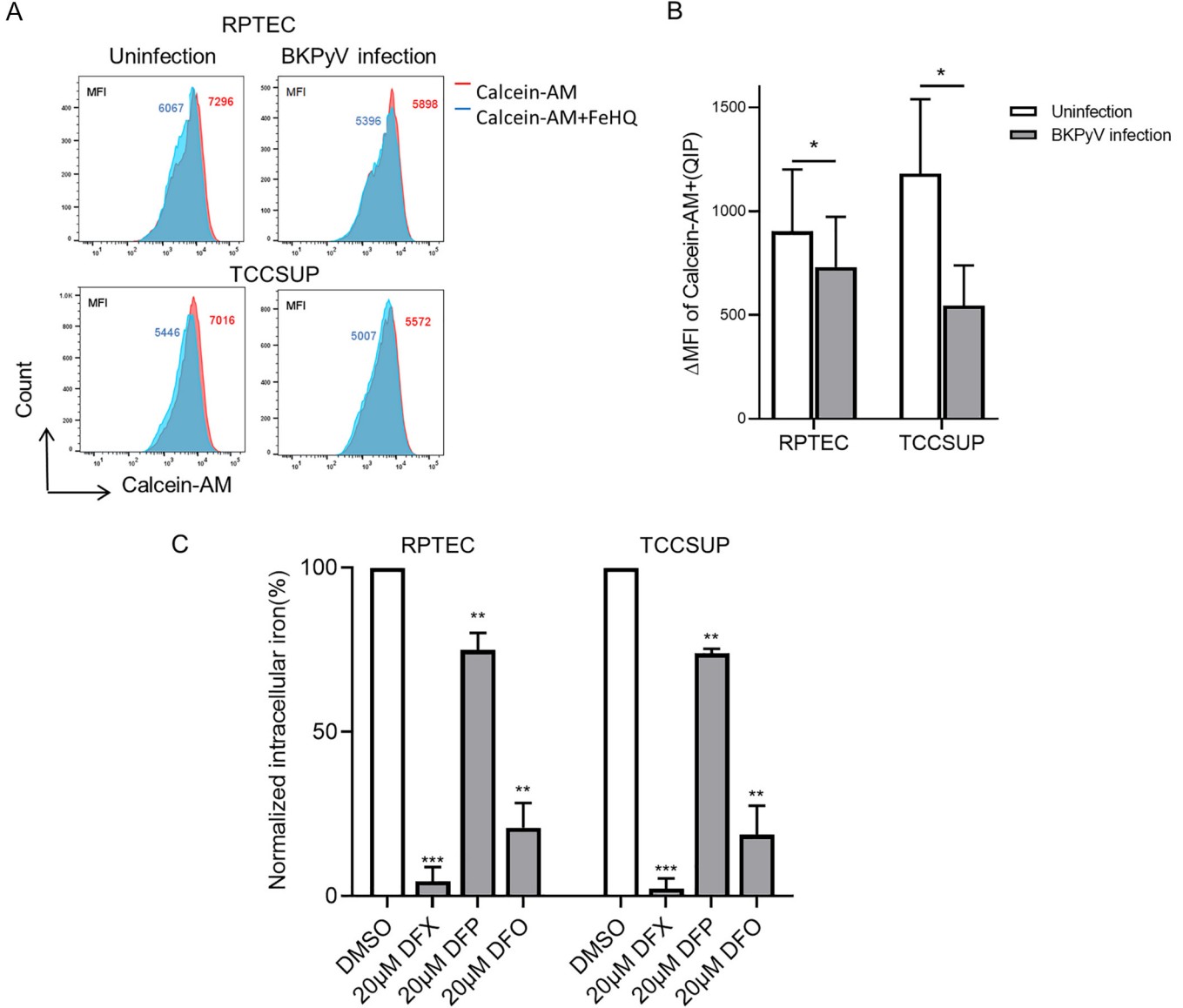

**FIG 1** Effect of BKPyV infection and iron chelators on the intracellular iron content in RPTECs and TCCSUP cells. (A) RPTECs and TCCSUP cells were infected with BKPyV or were left untreated. At 48 hpi, intracellular iron content was measured by flow cytometry assay based on calcein-AM fluorescence. (B) The average fluorescence intensity (MFI) of calcein-AM and the quenched iron pool (QIP) in BKPyV-infected/uninfected RPTECs and TCCSUP cells was calculated using Flowjo software. (C) RPTECs and TCCSUP cells were treated with DFX (20 $\mu$M), DFP (20 $\mu$M), and DFO (20 $\mu$M). At 72 hpi, intracellular iron content was measured by flow cytometry assay based on calcein-AM fluorescence and normalized to DMSO-treated cells. *, $P < 0.05$, **, $P < 0.005$, ***, $P < 0.0005$.

viral genome production as confirmed through qPCR analysis of BKPyV genome copy number (Fig. 2B).

The above data also showed that, at the same concentration, the inhibitory effects of the three iron chelators on virus infection were different: DFX and DFO were stronger than DFP at a concentration of 20 $\mu$M ($P < 0.05$). Consistent with these findings, we observed 98%, 81%, and 27% loss of virus infection in 20 $\mu$M DFX-, 20 $\mu$M DFA-, and 20 $\mu$M DFP-treated RPTECs, respectively (Fig. 2C and D). Considering the different ability of the three drugs to chelate iron, the virus inhibition may be related to the intracellular iron level. Therefore, we used calcein-AM staining to evaluate the levels of intracellular variable iron ions treated with three iron chelators. The results showed that three iron chelators significantly reduced intracellular labile iron content ($P < 0.05$; Fig. 1C). Moreover, the decreasing trend of iron content is basically consistent with the loss of virus infection (Fig. 1C; Fig. 2D). We preliminarily concluded that the inhibition rate of BKPyV was related to the degree of intracellular iron depletion.

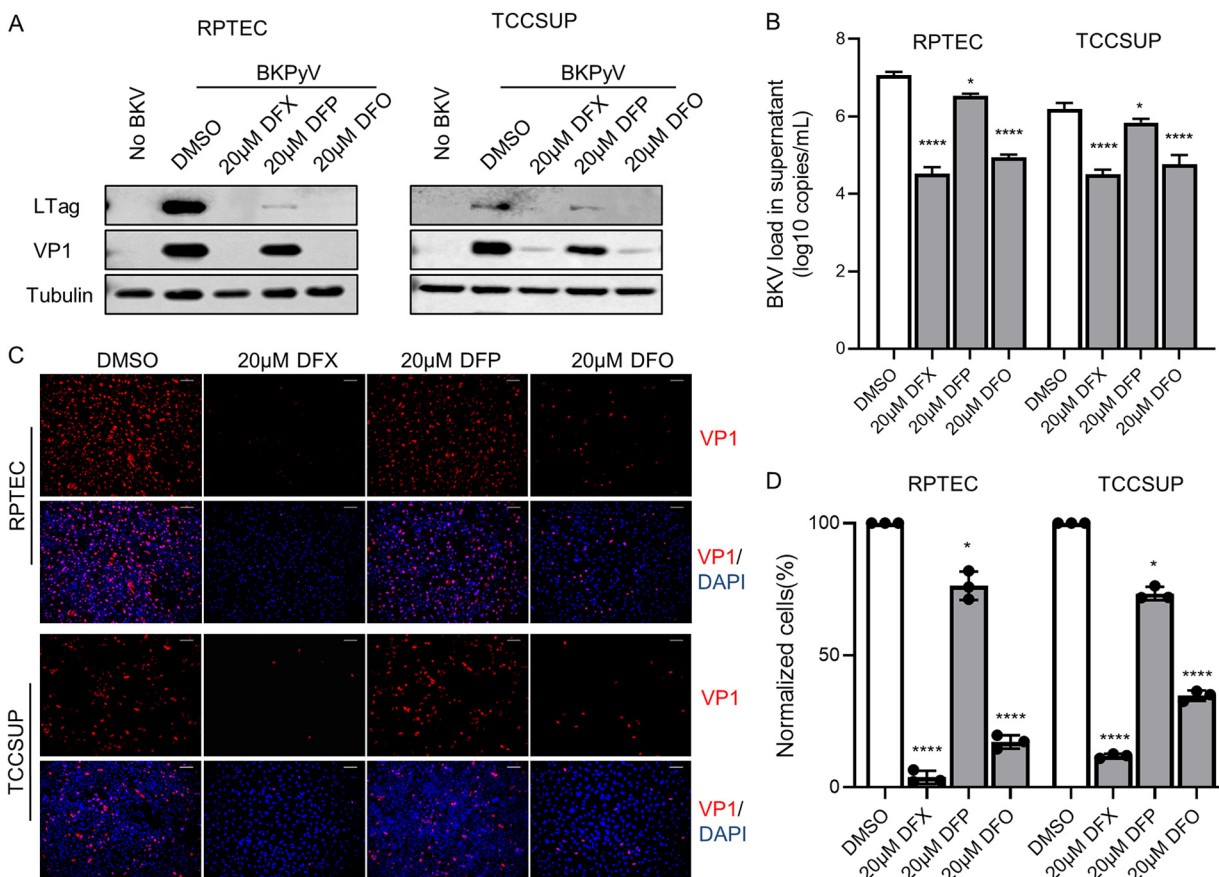

**FIG 2** Effect of iron chelators on BKPyV infection in RPTECs and TCCSUP cells. (A) BKPyV-infected RPTECs and TCCSUP cells were treated with DFX (20 $\mu$M), DFP (20 $\mu$M), and DFO (20 $\mu$M) for 72 h. Lysates from cells were resolved by SDS-PAGE and probed with anti-VP1, anti-LTag, and anti-beta-tubulin antibodies. Representative Western blots are shown. (B) Supernatants were harvested from BKV-infected RPTECs treated with DMSO, 20 $\mu$M DFX, 20 $\mu$M DFP, and 20 $\mu$M DFO, and BKPyV DNA loads were measured by qPCR. (C) At 72 hpi, BKPyV-infected RPTECs treated with DMSO, 20 $\mu$M DFX, 20 $\mu$M DFP, and 20 $\mu$M DFO from 2 hpi were fixed and stained with anti-VP1 antibody (red). Cell nuclei (blue) were stained with DAPI. The pictures were taken with a fluorescence microscope ($\times$10 objective). (D) The percentage of BKPyV-infected cells was quantified using Image J software and normalized to DMSO-treated cells. *, $P < 0.05$, ****, $P < 0.0001$.

Moreover, we also found that DFX and DFO in high concentrations (50 $\mu$M) inhibited cell proliferation ($P < 0.05$; Fig. S1E and F) but had only little effect on cell apoptosis (Fig. S1A to D).

**Intracellular irons are required for BKPyV infection.** To further explore the correlation between BKPyV infection and cellular iron levels, BKPyV infection assays were performed at a range of DFX or DFO concentrations (0 to 40 $\mu$M). With the increase of DFX or DFO concentration (0 to 10 $\mu$M), the protein expression of VP1 and VP3 and the viral load in the supernatant also showed a gradient decrease ($P < 0.05$; Fig. 3A to D; Fig. S2A to D). These results confirmed our previous inference that the BKPyV amplification was positively correlated with the cellular iron levels.

In order to further verify the inhibition of BKPyV infection caused by the decrease of cellular iron, an iron supplement assay was performed in BKPyV-infected cells. After treatment of BKPyV-infected RPTEC and TCCSUP cells with 5 $\mu$M DFX for 12 h, iron ions were supplemented by adding 0 to 20 $\mu$M ferriamine citrate (FAC). Iron supplementation could restore the protein expression of VP1 and VP3 and viral loads in the supernatant ($P < 0.05$; Fig. 3E to H). Moreover, within a certain range (0 to 5 $\mu$M), the viral protein expression and nucleic acid load increased with the supplemental iron concentration. This suggests that the inhibitory effect of iron chelators on BKPyV is dependent on cellular iron depletion. We also observed that when the supplemental FAC concentration reached 5 $\mu$M, the viral protein expression and nucleic acid load no longer increased synchronously. This may indicate that there was enough iron in the cell for

Microbiology
Spectrum

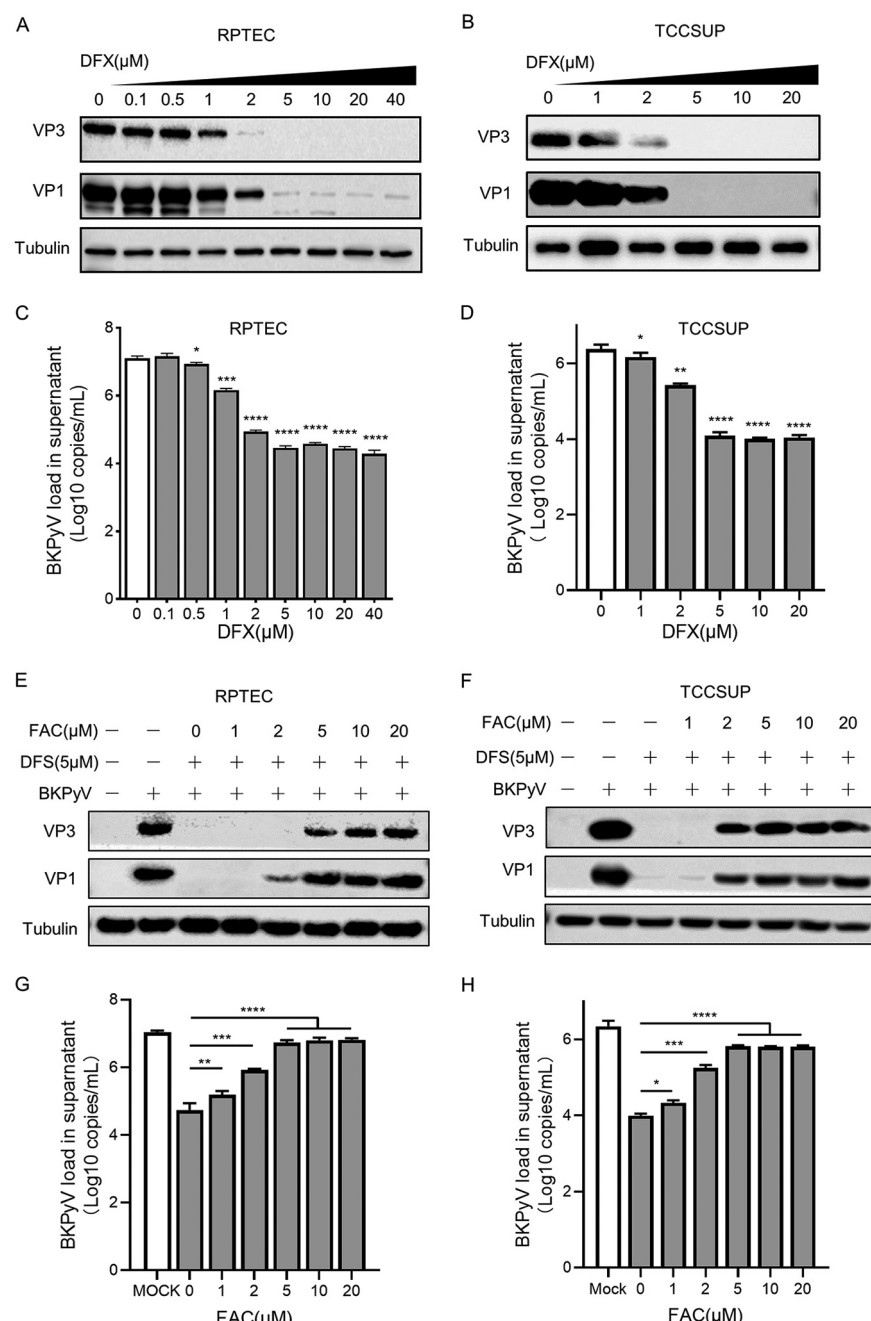

**FIG 3** Effect of decreasing intracellular iron concentrations on BKPyV proliferation in RPTECs and TCCSUP cells. (A and B) BKPyV-infected RPTECs and TCCSUP cells were treated with DFX at the indicated concentrations for 72 h. Lysates from cells were resolved by SDS-PAGE and probed with anti-VP1, anti-VP2/3, and anti-beta-tubulin antibodies. Representative Western blots are shown. (C and D) At 72 hpi, supernatants were harvested from BKPyV-infected RPTECs and TCCSUP cells treated with DFX at the indicated concentrations, and BKPyV DNA loads were measured by qPCR. (E and F) BKPyV-infected and DFX (5 $\mu$M)-treated RPTECs and TCCSUP cells were treated with FAC at the indicated concentrations from 12 h. At 72 h, lysates from cells were resolved by SDS-PAGE and probed with anti-VP1, anti-VP2/3, and anti-beta-tubulin antibodies. Representative Western blots are shown. (G and H) At 72 hpi, supernatants were harvested from RPTECs and TCCSUP cells, and BKPyV DNA loads were measured by qPCR. *, $P < 0.05$, **, $P < 0.005$, ***, $P < 0.0005$, ****, $P < 0.0001$.

the virus to proliferate. Taken together, these data identified a requirement for iron during the BKPyV life cycle in primary RPTE and TCCSUP cells.

**Iron depletion exerts an inhibitory effect after BKPyV enters the nucleus.** The stage of the BKPyV life cycle that requires the iron content was next investigated through time-of-addition experiments. Considering the strong anti-BKPyV effect and

less cell inhibition, 5 $\mu$M DFX was used as the representative iron-depleted drug. The RPTE cells were infected with BKPyV, and the 5 $\mu$M DFX was added at defined time points after infection. The addition of DFX at 2, 12, 24, and 32 h postinfection (hpi) resulted in decreased VP1 expression and viral loads ($P < 0.05$; Fig. 4A and B), thus suggesting that DFX could affect BKPyV even after a viral infection was established.

Next, we further clarified the specific window during which iron depletion acts to impair BKPyV. To successfully infect a cell, BKPyV needs to attach to the cell surface and interact with receptors to become internalized, which is accomplished by viral capsid proteins. In order to investigate whether iron depletion inhibited virus entry, a single-round BK pseudovirus (SrBKPV; Fig. S3A) infection assay was performed in cells with different iron levels. SrBKPV contains complete BKPyV capsid proteins and facilitates the simulation of the process before the entry of BKPyV into the nucleus. The infection rate is reflected by the expression of the enhanced green fluorescent protein (EGFP) or luciferase gene encased in the capsid. There was no significant reduction in the infection rate of two pseudoviruses in cells treated with 2 $\mu$M or 5 $\mu$M DFX compared with that in the untreated cells (Fig. 4C and D). Iron chelators did not affect the infection rate of SrBKPV, indicating that iron depletion had no inhibitory effect before BKPyV entered the nucleus.

To define whether iron depletion affects a series of processes after BKPyV enters the nucleus, the complete BKPyV genome was transfected into cells with different iron levels for replication and expression. Liposome-mediated transfection bypasses capsid-protein-dependent BKPyV entry and is therefore used to simulate the postnuclear entry process. The protein expressions of VP1 and VP3 and viral DNA replication were significantly inhibited by 5 $\mu$M DFX treatment and restored by 10 $\mu$M FAC ($P < 0.05$; Fig. 4E and F), suggesting that cellular iron depletion participates in inhibiting the proliferation of BKPyV after the virus enters the nucleus.

**Iron depletion inhibits the protein synthesis of BKPyV and exogenous gene.** To determine whether iron depletion inhibited the gene expression or genome replication of BKPyV, we constructed the pBKPyV-NCCR-EGFP plasmid based on the BKPyV genome and transfected it into cells with different iron levels (Fig. S3B). This plasmid preserves the complete sequence of NCCR and part of the early coding region of the BKPyV genome, which limits the regulation of LTag and STag, and replaces the late coding region with the EGFP gene sequence. In this experiment, the effect of progeny virus reinfection and LTag/STag regulation was excluded by transfection with pBKPyV-NCCR-EGFP in place of the complete BKPyV genome. The expression of EGFP in transfected cells with low iron levels was significantly inhibited ($P < 0.05$; Fig. 5A), thus suggesting that iron depletion exerts a dampening effect in the gene expression. However, the relative replication of the plasmid was not decreased in a low-iron environment (Fig. 5D), indicating that iron depletion did not inhibit the transfected gene replication. In addition, reverse transcriptase quantitative pPCR (RT-qPCR) showed that there was no significant difference in EGFP mRNA level between the two intracellular iron environments (Fig. 5C), which suggested that the inhibitory effect of low iron level on gene expression occurred after mRNA transcription. Meanwhile, transfection of another plasmid, pwb2B, which contains EF-1$\alpha$-activated VP1 protein, also showed that iron depletion inhibited EF-1$\alpha$-activated VP1 protein expression but did not change the relative loads of mRNA (Fig. 5B, E, and F). This suggests that iron depletion in cells may inhibit the expression of various exogenous genes, which is not directly related to the promoter of BKPyV. It is further verified that iron depletion has an inhibitory role after transcription of mRNA.

Subsequently, we examined the effect of iron depletion on BKPyV protein degradation in host cells after pretreatment with either the proteasome inhibitor MG132 or the autophagy inhibitor chloroquine (CQ). MG132 or CQ pretreatment did not abrogate the impairment of VP1 expression in iron-reduced cells (Fig. 5H), indicating that iron depletion does not promote the degradation of viral protein.

Given that iron depletion affected neither the transcription of the BKPyV gene nor its degradation of protein, we next tested whether iron depletion regulates the synthesis of

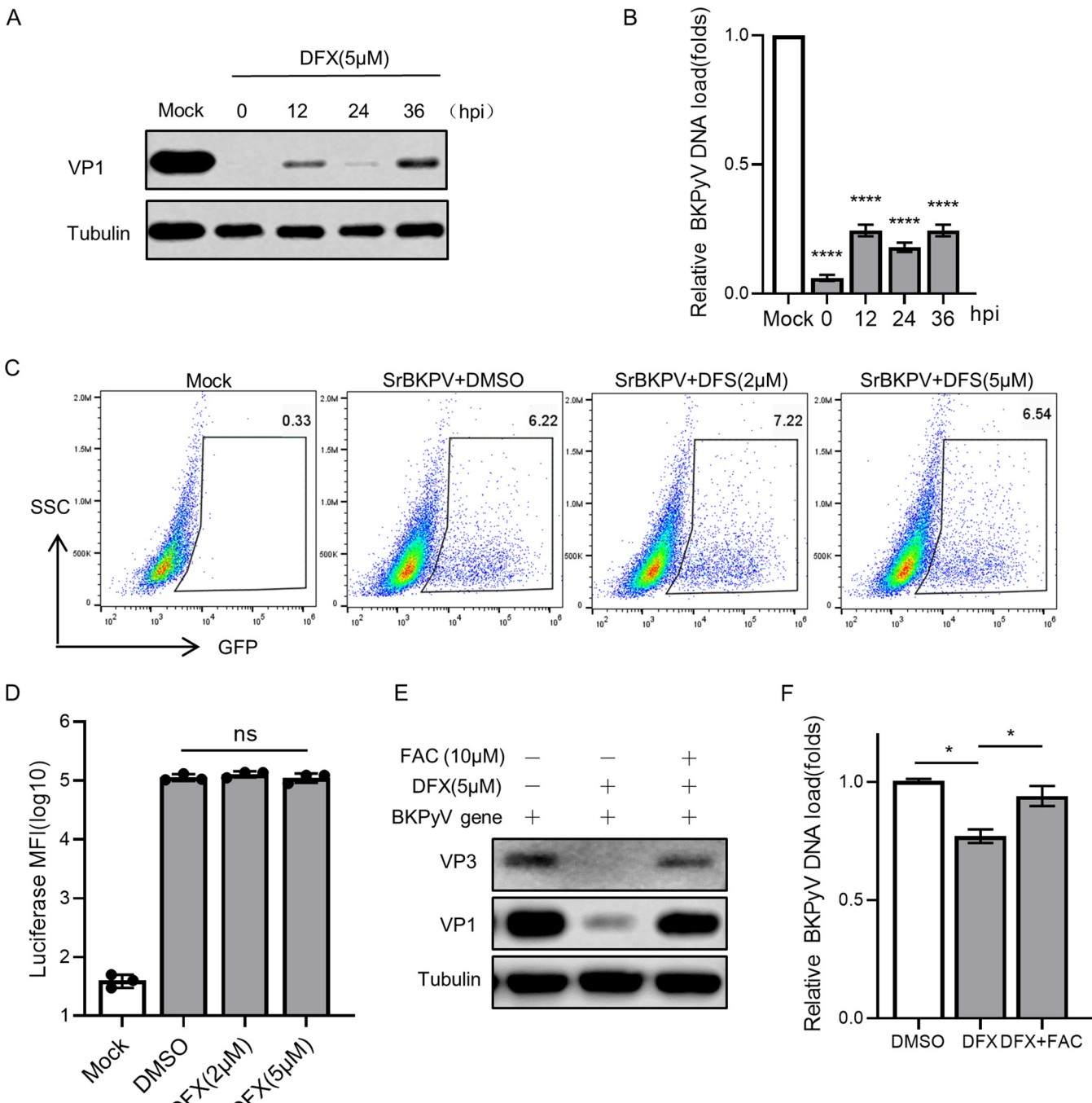

**FIG 4** Iron depletion exerts an inhibitory effect after BKPyV enters the nucleus. (A) BKPyV-infected RPTECs were treated with 5 $\mu$M DFX at 0, 12, 24, and 36 hpi, respectively. At 72 h, lysates from cells were resolved by SDS-PAGE and probed with anti-VP1 and anti-beta-tubulin antibodies. Representative Western blots are shown. (B) At 72 hpi, the DNA was extracted from BKPyV-infected RPTECs treated with 5 $\mu$M DFX at various time points, and BKPyV DNA loads were measured by qPCR. (C) RPTECs were infected with SrBKPV (EGFP) and treated with DMSO, DFX (2 $\mu$M), and DFX (5 $\mu$M) for 48 h. The EGFP positive cells were measured by flow cytometry. (D) RPTECs were infected with SrBKPV (luciferase) and treated with DMSO, DFX (2 $\mu$M), and DFX (5 $\mu$M) for 48 h. Lysates from cells were incubated with luciferin substrate, and mean fluorescence intensity was measured using a luminometer. (E) TCCSUP cells were transfected with a complete BKV genome and treated with DMSO and 5 $\mu$M DFX (and 10 $\mu$M FAC) for 48 h. Lysates from cells were resolved by SDS-PAGE and probed with anti-VP1, anti-VP2/3, and anti-beta-tubulin antibodies. Representative Western blots are shown. (F) DNA was extracted from transfected TCCSUP cells, and BKPyV DNA loads were measured by qPCR. ns, no significant differences, *, $P < 0.05$, ****, $P < 0.0001$.

viral proteins. Metabolic labeling with biotin was used to identify newly synthesized proteins and explore the changes in the expression of newly synthesized VP1 proteins in iron-reduced cells. The results showed that VP1 protein was highly expressed after the recovery of protein synthesis. The DFX treatment inhibited the amount of VP1 protein generation

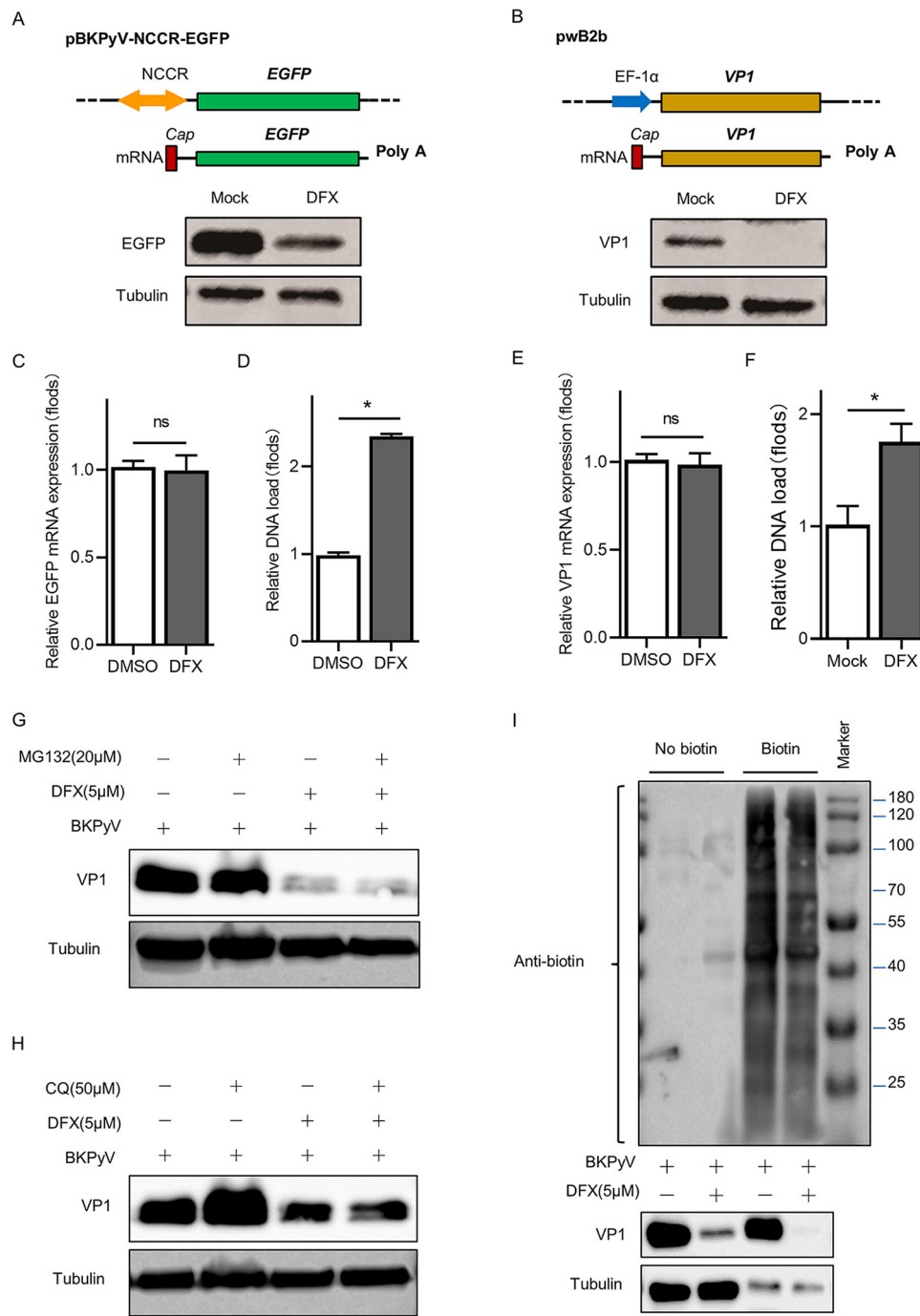

**FIG 5** Iron depletion inhibits the protein synthesis of BKPyV and exogenous genes. (A and B) TCCSUP cells were transfected with pBKPyV-NCCR-EGFP (A) or pwB2b (B) and treated with 5 μM DFX or DMSO for 48 h. Lysates from cells were resolved by SDS-PAGE and probed with anti-EGFP or anti-VP1 and anti-beta-tubulin antibodies. Representative Western blots are shown. (C and D) Total mRNA and DNA were extracted from TCCSUP cells transfected with pBKPyV-NCCR-EGFP. The EGFP mRNA expression (C) and DNA loads (D) were measured by RT-qPCR and qPCR, respectively. (E and F) Total mRNA and DNA were extracted from TCCSUP cells transfected with pwB2b. VP1 mRNA expression (E) and DNA loads (F) were measured by RT-qPCR and qPCR, respectively. (G and H) Immunoblot analysis of VP1 expression in lysates of 5 μM DFX-treated or untreated BKPyV-infected TCCSUP cells, treated with MG132 (20 μM) (G) or with CQ (50 μM) (H). (I) BKPyV-infected TCCSUP cells were treated with metabolic labeling reagents (ʟ-AHA). Subsequently, corresponding cell lysates were labeled with biotin for nascent proteins via azide/alkyne reaction, followed by biotin-avidin reaction and immunoblot analysis with anti-VP1, anti-beta-tubulin, and anti-biotin antibody. ns, no significant differences, *, $P < 0.05$.

($P < 0.05$) but had no significant effect on the expression of endogenous protein $\beta$-tubulin (Fig. 5I). In addition, the reduction of iron ions had a certain inhibitory effect on the synthesis of whole-cell proteins. Therefore, the decrease of iron ions could directly inhibit the synthesis of BKPyV protein, which is one of the mechanisms of iron depletion exerting the anti-BKPyV effect.

## DISCUSSION

This study evaluated the changes of intracellular iron in BKPyV-infected cells and explored the contribution of iron depletion to BKPyV amplification by creating a low-intracellular-iron environment. We demonstrated that iron-chelating-induced iron depletion inhibits BKPyV infection in primary renal tubular epithelial cells and bladder cancer cells. We also found that viral and exogenous protein synthesis was inhibited in cells with a low-iron environment.

Many viruses, such as HIV and HCMV, can direct iron use to their advantage by regulating iron transport during infection (18, 19). In this study, BK polyomavirus infection increased intracellular variable iron pool levels, suggesting that BKPyV may control intracellular iron availability by regulating iron transport or iron metabolism signaling pathways in the captive host cells. This result leads us to suspect that changes in the cellular iron environment may affect BKPyV infection. Therefore, we used iron chelators (drugs clinically used to treat hyperferremia) to create a low-iron environment in the cells to verify that the reduction of iron can inhibit the proliferation of BKPyV. Our data suggested that iron chelators may exhibit different anti-BKPyV effects due to their ability to chelate iron. In addition, the replication of BKPyV was restored by iron supplementation, which further indicated that the reproduction of BKPyV requires sufficient iron in cells and iron depletion is an effective therapy for inhibiting BKPyV infection.

According to the life cycle of BKPyV infection (20) and time-of-addition experiments, we discovered that the effect of iron depletion on BKPyV infection occurred in the late stage of infection. SrBKPV infection assay and BKPyV genomic transfection experiments further demonstrated this conclusion and confirmed that iron depletion has an inhibitory role after the virus enters the nucleus but not before. This phased experimental design is more reasonable and effective for studying the specific action period of iron depletion and is also suitable for exploring other anti-BKPyV drugs. After the BKPyV genome enters the nucleus, it goes through a series of processes, including viral gene expression, gene replication, viral assembly, and exocytosis (12). Thus, we verified the specific stage of the effect of iron depletion on which BKPyV infection occurred. In order to simulate the NCCR-mediated protein expression and DNA replication of BKPyV and to eliminate the interference of progeny virus reinfection and the regulatory effects of early protein LTag and STag, we designed a plasmid, pBKPyV-NCCR-EGFP. By transfection of this plasmid, we verified that a low-iron environment could inhibit the expression of the BKPyV gene and exogenous genes. At the same time, the expression of VP1 in plasmid pwB2b was inhibited in a low-iron environment, suggesting that iron depletion may inhibit the expression of a variety of exogenous genes. Despite the different entry modes, the BKPyV genome and the transfected exogenous genes have very similar expression and replication processes. Both of them utilize cellular machinery for their own protein expression, so the mechanism of BKPyV and exogenous gene expression inhibition in iron-deficient cells is consistent. This inhibitory effect may also apply to other DNA viruses that replicate on host cells. We also concluded that this inhibition effect might not directly affect promoter function because the two plasmids have different promoters but suffer the same inhibition results. Of course, this may vary as the expression inhibition effect of exogenous genes containing other promoters or other viral genes is not overridden. In the study, we also observed that while the gene expression was inhibited, the total amount of DNA and mRNA did not have the same reduction trend, thus proving that the inhibition of the expression of an exogenous gene and BKPyV gene occurred after transcription. This is consistent with the study of Zhu et al. (21), who suggested that iron depletion is

involved in posttranscriptional regulation. BKPyV does not encode RNA polymerase, and all other gene expression factors are supplied by the host cell. Therefore, it is possible for iron depletion to inhibit the synthesis of viral proteins through cellular metabolism directly. We confirmed this hypothesis with metabolic labeling experiments, in which viral protein synthesis was significantly inhibited in an intracellular low-iron environment. At the same time, we also ruled out the effect of iron depletion on protein degradation by adding protease inhibitors. Although viral proteins were largely inhibited, global protein synthesis was reduced only slightly, and endogenous reference proteins were not significantly changed. In the infected cells, BKPyV proliferates by synthesizing large amounts of viral proteins during activation, and the transfected plasmids also express large amounts of proteins in a short period. This abnormally large amount of protein synthesis in cells is restricted by iron levels. Therefore, iron depletion can selectively inhibit the expression of a large number of nonautologous proteins in cells, which may be an important reason for the inhibition of BKPyV proliferation in a low-iron environment.

Previous studies have reported that iron chelators inhibit the infection of a variety of viruses through various mechanisms (13). Here, we reported the inhibitory effect of a low-iron environment on BKPyV protein synthesis, which provides new insights for understanding the correlation between BKPyV and iron environment in host cells and new ideas for clinical treatment of BKPyV. For patients with BKPyV infection combined with iron overload, iron depletion therapy is undoubtedly a very appropriate treatment, which can simultaneously solve the symptoms of iron overload and infection. However, possible adverse reactions should also be considered since some patients with BKPyV infection after kidney transplantation are accompanied by anemia symptoms. On the one hand, we can further explore more precise correlations between iron levels and virus suppression and monitor iron levels to keep the body safe while suppressing virus proliferation. On the other hand, taking iron regulatory protein as a new research direction may lead toward the exploration of more precise mechanisms of action and identification of more effective anti-BKPyV targets, which are worthy of further study.

In conclusion, we demonstrated that iron depletion inhibited BKPyV infection in an *in vitro* infection model, possibly through selective inhibition of extracellular protein synthesis in low-iron environments. Further exploration of the target proteins of iron-regulating viral infection could have great potential in developing new strategies for urgently needed anti-BKPyV therapies.

## MATERIALS AND METHODS

**Cell culture.** Human primary renal proximal tubule epithelial cells (RPTEC; ATCC) were maintained in renal epithelial cell growth medium with fetal bovine serum (FBS) and epithelial cell growth supplement-animal (ready-to-use) (Sciencell). Human bladder carcinoma cells (TCCSUP; HTB-5, ATCC) were cultured in minimal essential medium (MEM) (Gibco Life Technologies) supplemented with 10% fetal bovine serum (FBS; Gibco Life Technologies). All cells were cultured in a humidified atmosphere with 5% $CO_2$ at 37°C.

**BKPyV infections and drug treatments.** BKPyV stocks were initially propagated in Vero cells from viruses obtained from ATCC (VR-837). RPTECs or TCCSUP cells seeded into 6-well plates ($2 \times 10^5$ cells per well) were incubated with BKPyV for 2 h before surplus viruses were removed. BKPyV was added with a multiplicity of infection (MOI) of 1, unless indicated otherwise. The cells were washed once with phosphate-buffered saline (PBS; Corning), and the complete medium with or without drugs was added. A dimethyl sulfoxide (DMSO) control at a corresponding concentration was included in the experiments. The following drugs were used in this study: deferasirox (DFX; APExBIO, A8639), 3-hydroxy-1,2-dimethyl-4(1H)-pyridone (DFP; Aladdin, H122577), deferoxamine mesylate (DFO; Aladdin, D302525), ferriamine citrate (FAC; Sangon Biotech, A500061), MG132 (Sangon Biotech, T510313), and chloroquine diphosphate salt (CQ; Sigma, C6628).

**Time-of-addition experiments.** RPTECs in 6-well plates ($2 \times 10^5$ cells per well) were incubated with BKPyV for 2 h. After the surplus virus was removed, the cells were washed in phosphate-buffered saline (PBS) and treated with 5 $\mu$M DFX at 0, 2, 12, 24, and 36 h postinfection (hpi), respectively. Cells were then incubated for 72 h and then collected for subsequent detection.

**SrBKPV production and infection.** Production of SrBKPV was performed as described in Schowalter et al. 2011 (22). The capsid protein expression plasmids pwB2b (Addgene, no. 32094) and pwB3b (Addgene, no. 32106), a gift from Christopher Buck, were cotransfected with the reporter plasmid pEGFP-N1 (Clontech) or pGL3-luciferase (Promega), respectively. After 48 h, the cells were harvested and lysed in Dulbecco's phosphate-buffered saline (DPBS; Corning) supplemented with 9.5 mM $MgCl_2$, 25 mM ammonium sulfate (Sigma), 0.5% Triton X-100 (Sigma), and 0.1% RNase A/T1 cocktail (Thermo

Scientific). The cell lysate was incubated at 37°C overnight with the goal of promoting capsid maturation. Lysates containing mature capsids were clarified by centrifugation for 10 min at 5,000 × g twice. The virus particles were purified by centrifugation at 234,000 × g for 3.5 h with 25% sucrose as the bedding material and then released in 100 $\mu$l Opti-MEM medium (Corning).

For SrBKPV infection, cells in 96-well plates (5 × 10³ cells per well) were incubated with 2 $\mu$l SrBKPV for 2 h. After the surplus virus was removed, the cells were washed once with PBS and incubated with the medium with or without iron chelators for 48 h. Subsequently, green fluorescent protein (GFP)-positive cells were detected with flow cytometry (BD), and luciferase activity was detected by luciferase reporter assay kit (YEASEN) according to the manufacturer's instructions.

**BKPyV genome amplification.** The complete BKPyV genome was digested by BamHIenzyme (TaKaRa) and linked to the pBlunt vector to construct the pBlunt-BKPyV plasmid. Then, the pBlunt-BKPyV plasmids were transformed into *Escherichia coli* for amplification. After amplification, the pBlunt-BKPyV plasmids were extracted from *E. coli* using a plasmid DNA purification kit (Macherey-Nagel). The extracted pBlunt-BKPyV plasmids were digested with BamHI enzyme and subjected to agarose electrophoresis. A large amount of linearized BKPyV genomic DNA (about 5,000 bp) was cut out and recovered using a Nucleospin gel cleanup kit (Macherey-Nagel). The augmented proliferative BKPyV genomes were obtained by linear BKPyV genomic DNA self-linking via T4 ligase (TaKaRa).

**pBKPyV-NCCR-EGFP production.** Sequences from truncated STag CDS to the end of NCCR late promoter of BKPyV were obtained by PCR using pBlunt-BKPyV (Archtype NCCR) plasmid as a template, and NotI and XhoI restriction sites were added at the terminus, respectively. To construct pGL3-NCCR-luciferase, the pGL3 BASIC vector was cleaved with NotI and XhoI enzyme and then ligated with the target fragment. Then, the luciferase sequence in the above plasmid was replaced by the EGFP sequence of pEGFP-N1 through the ligation of XhoI and XbaI restriction sites to product pGL3-NCCR-EGFP (pBKPyV-NCCR-EGFP) plasmid.

**DNA transfection.** Transfection assays were performed when TCCSUP cells were grown to 70% confluence in complete MEM. Transfection of BKPyV genome and plasmid was performed using FuGENE HD transfection reagent (Promega) according to the manufacturer's instruction. Cells were collected 48 h after transfection.

**Metabolic labeling.** TCCSUP cells were cultured in 6-well plates (2 × 10⁵ cells per well) and infected with BKPyV (MOI 1) for 12 h. To stop protein synthesis in the cells, the complete MEM medium was removed, and the cells were cultured in a methionine-free Dulbecco's modified Eagle's medium (DMEM) (Gibco Life Technologies) containing 10% FBS with or without 5 $\mu$M DFX. After 1 h, 20 $\mu$M L-azidohomoalanine (L-AHA; MCE) was added into the cell culture supernatant to restore protein synthesis. The cells were then cultured for 8 h and collected.

**Newly synthetic proteins pulldown.** L-AHA-labeled cells were lysed in radioimmunoprecipitation assay (RIPA) buffer (Biosharp) containing protease inhibitor cocktail (TaKaRa). Proteins from cell lysis were collected and quantified using the bicinchoninic acid (BCA) protein quantification kit (Biosharp), and the protein concentration was adjusted. By clicking chemical reactions, proteins containing L-AHA were labeled using BPDO-PEG4000-biotin (MCE) with a final concentration of 40 $\mu$M. Excess streptavidin magnetic beads (YEASEN) were added to the protein mixture for adsorption of biotin-labeled proteins, and the unbound proteins were washed away with a RIPA buffer. The remaining beads and proteins were added with 40 $\mu$l 4× SDS-PAGE loading buffer (TaKaRa) containing 2 mM BPDO-PEG4000-biotin and incubated at 98°C for 10 min. Samples were stored at −80°C for subsequent detection.

**Measurement of intracellular iron.** Measurement of intracellular iron was performed as described previously (23). RPTECs or TCCSUP cells were inoculated in 6-well plates (2 × 10⁵ cells per well) with or without BKPyV infection (and/or iron chelator treatment). Cells were washed twice with PBS followed by incubation with 1 mM calcein acetoxymethyl ester (calcein-AM; Aladdin) in PBS for 15 min. After staining, cells were washed twice with PBS and trypsinized to a single-cell suspension. All cell samples were divided evenly into two centrifuges: one was stored in PBS, and the other was coincubated with FeHQ buffer (5 $\mu$M ferrous chloride and 10 $\mu$M 8-hydroxyquinoline in PBS buffer) for 30 min. The excess FeHQ buffer was washed off with PBS buffer. The cells were resuspended with PBS and analyzed by flow cytometry. The average fluorescence intensity (MFI) of calcein-AM and the quenched iron pool (QIP) were calculated as follows: QIP = untreated MFI(calcein-AM) − FeHQ-treated MFI(calcein-AM). Intracellular iron was negatively correlated with QIP.

**Western blotting.** Western blotting was carried out as described previously (24). Cells in 6-well plates were first solubilized in 100 $\mu$M RIPA lysate containing protease inhibitor cocktail. Proteins were then separated by SDS-polyacrylamide gel electrophoresis (PAGE), blotted onto a polyvinylidene difluoride (PVDF) membrane, and detected using an infrared fluorescent detection system (Bio-RAD). The following primary antibodies were used: SV40 large T antigen rabbit monoclonal antibody (D1E9E; CST), BKPyV VP1 monoclonal antibody (3B2; Abnova), anti-SV40 VP2+VP3 monoclonal antibody (Abcam), and horseradish peroxidase (HRP)-conjugated beta-tubulin antibody (Genetex).

**Quantitative PCRs and RT-qPCRs.** Total RNA was extracted using the E.Z.N.A. total RNA kit I (Macherey-Nagel) according to the manufacturer's protocol. The RNA concentrations were measured by using a NanoDrop apparatus. RNA (1 $\mu$g) removed contaminated DNA was reverse transcribed using PrimeScript RT reagent kit with gDNA eraser (TaKaRa). Intracellular DNA was extracted using the DNAiso reagent (TaKaRa) after extensive cell washing. Quantitative PCRs (qPCRs) were performed on a CFX96 real-time PCR detection system (Bio-RAD) using the TB green premix *Ex Taq* (TAKAR). Primers were as follows (from 5′ to 3′): BKPyV VP1 (F: AGTGGATGGGCAGCCTATGTA; R: TCATATCTGGGTCCCCTGGA); GFP (F: GAACGGCATCAAGGTGAACT; R: TGCTCAGGTAGTGGTTGTCG); pwB2b VP1(F: GGTCCCCAAGTTGCTGATCA; R: CGGGTGTTCTCGTTCTTGGA); actin (F: TTTTCACGGTTGGCCTTAGG; R: AAGATCTGGCACCACACCTTCT).

**Quantification of extracellular BKPyV loads.** Cell culture supernatants were harvested and stored at −80°C until the extracellular BKPyV DNA load was quantified by qPCR using primers and a probe targeting the BKPyV LTag gene. qPCRs were performed using the premix *Ex Taq* (probe TaKaRa). Primers and probe for LTag were as follows: forward 5′-AGCAGGCAAGGGTTCTATTACTAAAT-3′; reverse 5′-GAAGCAACAGCAGATTC TCAACA-3′; probe: 5′-FAM-AAGACCCTAAAGACTTTCCCTCTGATCTACACCAGTTT-TAMRA-3′

**Immunofluorescence staining and microscopy.** Immunofluorescence staining was performed as described previously (24). Briefly, cell slides were fixed in 4% paraformaldehyde (PFA; YEASEN) for 10 min and permeabilized with 0.1% Triton X-100 for 15 min at room temperature (RT). Fixed cells were blocked with 5% bovine serum albumin (BSA; Sigma-Aldrich) for 20 min. Primary and secondary antibodies were diluted in 2% BSA and incubated at room temperature for 50 min each. The primary antibody was the BKPyV VP1 monoclonal antibody (3B2; Abnova), and the secondary antibody was anti-mouse IgG-Alexa Fl$\mu$or 647 (Abcam). DNA was labeled with DAPI (4′,6-diamidino-2-phenylindole) dye (CST). Fluorescence was detected using an inverted fluorescence microscope (EVOS XL Core, Thermo Fisher Scientific). All images were processed using ImageJ software (NIH, Bethesda, MD, USA).

**Statistical analysis.** When appropriate, the mean and standard deviation (SD) were calculated, and $P$ values were determined using the unpaired $t$ test (GraphPad). A $P$ value of <0.05 was considered to be statistically significant.

## SUPPLEMENTAL MATERIAL

Supplemental material is available online only.
**SUPPLEMENTAL FILE 1**, PDF file, 0.8 MB.

## ACKNOWLEDGMENTS

We thank Nannan Wu at the Shanghai Public Health Clinical Center of Fudan University and Chao Hu at the Zhongshan Hospital of Fudan University for critical and helpful comments.

This study was supported by the Shanghai Shenkang Hospital Development Center Clinical Science and Technology Innovation Project (grants number SHDC12018101 to T.Z.), the National Natural Science Foundation of China (grants number 81873621 to T.Z.), the Shanghai Municipal Health Commission Scientific Research Project (grant number 20194Y0088 to Y.Z.), the Jinshan District Science and Technology Committee Fund of Shanghai (grant number 2020-3-64 to G.W.), and the Shanghai Municipal Health Commission Scientific Research Project (grant number 21S11902600 to G.W.).

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
