## [Reviewer comments · Microbiology Spectrum]

Microbiology Spectrum

Intracellular low iron exerts anti-BK polyomavirus effect by inhibiting the protein synthesis of exogenous genes

Jiajia Sun, Yejing Shi, Yumin Hou, Chunlan Hu, Yigang Zeng, Guoyi Wu, and Tongyu Zhu

Corresponding Author(s): Tongyu Zhu, Shanghai Public Health Clinical Center, Fudan University

Review Timeline:

Submission Date:	July 30, 2021
Editorial Decision:	September 7, 2021
Revision Received:	September 16, 2021
Accepted:	October 14, 2021

Editor: Clinton Jones

Reviewer(s): The reviewers have opted to remain anonymous.

Transaction Report:

DOI: <https://doi.org/10.1128/Spectrum.01094-21>

September 7, 2021

Dr. Tongyu Zhu
Shanghai Public Health Clinical Center, Fudan University
Shanghai
China

Re: Spectrum01094-21 (Intracellular low iron exerts anti-BK polyomavirus effect by inhibiting the protein synthesis of exogenous genes)

Dear Dr. Tongyu Zhu:

Thank you for submitting your manuscript to Microbiology Spectrum. When submitting the revised version of your paper, please provide (1) point-by-point responses to the issues raised by the reviewers as file type "Response to Reviewers," not in your cover letter, and (2) a PDF file that indicates the changes from the original submission (by highlighting or underlining the changes) as file type "Marked Up Manuscript - For Review Only". Please use this link to submit your revised manuscript - we strongly recommend that you submit your paper within the next 60 days or reach out to me. Detailed information on submitting your revised paper are below.

Link Not Available

Sincerely,

Clinton Jones

Journals Department
Reviewer comments:

Reviewer #1 (Comments for the Author):

Title: Intracellular low iron exerts anti- BK polyomavirus effect by inhibiting the protein synthesis of exogenous genes

Study Summary: The study uses an in vitro model (renal tubular epithelial and urinary bladder cancer cells) to study how drug-dependent iron depletion hampers BK polyomavirus (BKV) replication. After determining that BKV replication is iron-release dependent the authors go into a lot of detail to study the effects of protein, RNA, viral genome and promoter regulation depending on drug-induced iron availability. The study suggests that iron depletion exerts a restricting effect in viral gene expression (post-mRNA transcription), after examining cell entry, replication, etc.

Overall Impression: It is a well-done study that uses a wide arrange of techniques to analyze the effect of drug-induced iron depletion on BKV replication.

Major Comments:

Line 72 and 83: "Therefore, understanding the relationship between BKPyV and the host cell environment is critical for the

development of anti-BKPyV therapeutics." and "However, a requirement for an intracellular iron environment and its contribution to BKPyV infection and replication has not yet been explored." The authors imply that using iron-depleting drugs would be a BKV-specific treatment, which they cite themselves is not necessarily true as that inhibits viral replication among other viruses as well.

Line 97: Methods on drug treatments. Drug concentrations are not referenced. More information on how (concentration and frequency) the drugs are currently prescribed in clinical settings and how the concentrations used in the manuscript would mirror that (or not). Hence, what is the clinical relevance of the concentrations used? Line 112: "After the surplus virus was removed, the cells were washed in PBS and treated with 5 μ M DFX at 0, 2, 12, 24, and 36 h post-infection (hpi), respectively." The first set of experiments were done using 20 μ M?

The authors start with 3 drugs, and proceed only with DFX without discussing why or the used concentrations. It is clear that DFX has the highest iron-chelating effect but also seems to 20 μ M kills cells (Fig.2C, few nuclei in IFA?). Are there any possible explanation of varying degrees of effect since all drugs were used at same concentration in the beginning? Line 351: "Our data suggested that iron chelators exhibited different anti-BKPyV effects due to their ability to chelate iron..." while there is a dose-dependent effect its not clear all drugs worked the same.

The authors use BKPyV and BKV interchangeably. Should be streamlined.

Figure 2 uses LTag and VP1 in Western Blot and Figure 3 used VP1 and VP3 in Western Blot. Is there a specific reason to use those sets of structural and non-structural proteins?

Line 391: Captive cells = infected cells?

Line 401: "The next step should be to examine the anti-BKPyV effect of iron chelators in vivo." Use of clinically relevant concentrations and further studies on physiological effect of in vitro models first?

Line 402: "Moreover, possible adverse reactions should also be considered since patients with BKPyV infection after kidney transplantation are often accompanied by anemia symptoms." This is the only sentence mentioning any possible clinical side-effects of using iron-chelators in clinical setting, without discussing current clinical applications (except that its currently used for hyperferremia).

Minor Comments:

Line 66: TAGs = Tags, to be consistent

Line 76 and 77: Add references

Line 93: Human bladder carcinoma cells (TCCSUP; HTB-5, ATCC)

Line 242: the expression of early LTag protein...

Line 357: depletion on BKPyV infection occurred

Reviewer #2 (Comments for the Author):

The manuscript, Intracellular low iron exerts anti- BK polyomavirus effect by inhibiting the protein synthesis of exogenous genes, by the Zhu lab is a fairly comprehensive, rigorous, and unique study of the role cellular iron plays during a virus infection. They tested three drugs in clinical use for their ability to chelate intracellular iron and correlated this with inhibition of virus gene expression and subsequent viral loads in tissue culture supernatant. The results were impressive with each drug inhibiting infection between 27 and 98%. This correlated with drugs ability to reduce intracellular iron. The effects could be reversed by providing iron exogenously.

Major concern:

They provide no data regarding the potential cytotoxic or cytostatic effects of these drugs on the cell. These controls would be critical for proper interpretation of the results.

Summary:

This is an important and unique study that could lead to promising treatments for polyomavirus associated diseases. The experiments are well-controlled, with the exception of controls for cytotoxic/cytostatic effects, and the conclusions are supported by the data.

Staff Comments:

Preparing Revision Guidelines

Please return the manuscript within 60 days; if you cannot complete the modification within this time period, please contact me. If you do not wish to modify the manuscript and prefer to submit it to another journal, please notify me of your decision immediately so that the manuscript may be formally withdrawn from consideration by Microbiology Spectrum.

Title: Intracellular low iron exerts anti- BK polyomavirus effect by inhibiting the protein synthesis of exogenous genes

Study Summary: The study uses an in vitro model (renal tubular epithelial and urinary bladder cancer cells) to study how drug-dependent iron depletion hampers BK polyomavirus (BKV) replication. After determining that BKV replication is iron-release dependent the authors go into a lot of detail to study the effects of protein, RNA, viral genome and promoter regulation depending on drug-induced iron availability. The study suggests that iron depletion exerts a restricting effect in viral gene expression (post-mRNA transcription), after examining cell entry, replication, etc.

Overall Impression: It is a well-done study that uses a wide arrange of techniques to analyze the effect of drug-induced iron depletion on BKV replication.

Major Comments:

Line 72 and 83: “Therefore, understanding the relationship between BKPyV and the host cell environment is critical for the development of anti-BKPyV therapeutics.” and “However, a requirement for an intracellular iron environment and its contribution to BKPyV infection and replication has not yet been explored.” The authors imply that using iron-depleting drugs would be a BKV-specific treatment, which they cite themselves is not necessarily true as that inhibits viral replication among other viruses as well.

Line 97: Methods on drug treatments. Drug concentrations are not referenced. More information on how (concentration and frequency) the drugs are currently prescribed in clinical settings and how the concentrations used in the manuscript would mirror that (or not). Hence, what is the clinical relevance of the concentrations used? Line 112: “After the surplus virus was removed, the cells were washed in PBS and treated with 5 μ M DFX at 0, 2, 12, 24, and 36 h post-infection (hpi), respectively.” The first set of experiments were done using 20 μ M?

The authors start with 3 drugs, and proceed only with DFX without discussing why or the used concentrations. It is clear that DFX has the highest iron-chelating effect but also seems to 20 μ M kills cells (Fig.2C, few nuclei in IFA?). Are there any possible explanation of varying degrees of effect since all drugs were used at same concentration in the beginning? Line 351: “Our data suggested that iron chelators exhibited different anti-BKPyV effects due to their ability to chelate iron...” while there is a dose-dependent effect its not clear all drugs worked the same.

The authors use BKPyV and BKV interchangeably. Should be streamlined.

Figure 2 uses LTag and VP1 in Western Blot and Figure 3 used VP1 and VP3 in Western Blot. Is there a specific reason to use those sets of structural and non-structural proteins?

Line 391: Captive cells = infected cells?

Line 401: “The next step should be to examine the anti-BKPyV effect of iron chelators in vivo.”
Use of clinically relevant concentrations and further studies on physiological effect of in vitro models first?

Line 402: “Moreover, possible adverse reactions should also be considered since patients with BKPyV infection after kidney transplantation are often accompanied by anemia symptoms.”
This is the only sentence mentioning any possible clinical side-effects of using iron-chelators in clinical setting, without discussing current clinical applications (except that its currently used for hyperferremia).

Minor Comments:

Line 66: TAGs = Tags, to be consistent

Line 76 and 77: Add references

Line 93: Human bladder carcinoma cells (TCCSUP; HTB-5, ATCC)

Line 242: the expression of early LTag protein...

Line 357: depletion on BKPyV infection occurred

Reviewer comments:

Reviewer #1 (Comments for the Author):

Title: Intracellular low iron exerts anti- BK polyomavirus effect by inhibiting the protein synthesis of exogenous genes

Study Summary: The study uses an in vitro model (renal tubular epithelial and urinary bladder cancer cells) to study how drug-dependent iron depletion hampers BK polyomavirus (BKV) replication. After determining that BKV replication is iron-release dependent the authors go into a lot of detail to study the effects of protein, RNA, viral genome and promoter regulation depending on drug-induced iron availability. The study suggests that iron depletion exerts a restricting effect in viral gene expression (post-mRNA transcription), after examining cell entry, replication, etc.

Overall Impression: It is a well-done study that uses a wide arrange of techniques to analyze the effect of drug-induced iron depletion on BKV replication.

Major Comments:

Line 72 and 83: "Therefore, understanding the relationship between BKPyV and the host cell environment is critical for the development of anti-BKPyV therapeutics." and "However, a requirement for an intracellular iron environment and its contribution to BKPyV infection and replication has not yet been explored." The authors imply that using iron-depleting drugs would be a BKV-specific treatment, which they cite themselves is not necessarily true as that inhibits viral replication among other viruses as well.

Reply: Thank you for your rigorous consideration. In fact, we intended to describe the importance of the host cell environment for BKPyV infection and to introduce the idea of studying the correlation between BKPyV and intracellular iron. Our expression in the manuscript was not rigorous, and we have modified.

Line 97: Methods on drug treatments. Drug concentrations are not referenced. More information on how (concentration and frequency) the drugs are currently prescribed in clinical settings and how the concentrations used in the manuscript would mirror that (or not). Hence, what is the clinical relevance of the concentrations used? Line 112: "After the surplus virus was removed, the cells were washed in PBS and treated with 5 μ M DFX at 0, 2, 12, 24, and 36 h post-infection (hpi), respectively." The first set of experiments were done using 20 μ M?

Reply: Thank you for your rigorous consideration. As clinically approved drugs, iron chelators are limited to the treatment of iron overload. However, there is no uniform standard for blood concentration of drugs and no data on the clinical treatment of viral infections by these drugs. In this paper, we refer to the concentration of drugs used to inhibit HIV infection in vitro in previous literature (*Virology* 367 (2007) 324-333; *The Journal of Infectious Diseases* 2000;181:484–90). Based on above research, we decided to use 20 μ M drugs (24h subcutaneous infusion of 100 mg/kg DFO in patient with iron overload) as a preliminary demonstration of the results of "Iron chelators inhibit BKPyV infection", because the drug has little effect on cell proliferation at this concentration.

In subsequent dose-inhibition assay, we further found that 5 μ M was the lowest concentration at which DFX exerted strongest inhibitory effect (as shown in Figure 3A-D), so we considered 5 μ M DFX was a more reasonable choice in time-inhibition experiments and studies on mechanism. Sorry for not explaining this point, we have added it in the fourth part of result.

The authors start with 3 drugs, and proceed only with DFX without discussing why or the used concentrations. It is clear that DFX has the highest iron-chelating effect but also seems to 20 μ M kills cells (Fig.2C, few nuclei in IFA?). Are there any possible explanation of varying degrees of effect since all drugs were used at same concentration in the beginning? Line 351: "Our data suggested that iron chelators exhibited different anti-BKPyV effects due to their ability to chelate iron..." while there is a dose-dependent effect its not clear all drugs worked the same.

Reply: Thank you for your rigorous consideration. We demonstrated that the iron chelators play an anti-BKPyV role through iron depletion by using iron supplement assay (Figure 3E-H). As the reviewer said, DFX has the highest iron-chelating effect, and we verified DFX had the strongest anti-BKPyV effect at the same concentration (Figure 2A-D). Therefore, DFX was used as a representative drug to deplete iron in the subsequent study on mechanism. We have added explain in the fourth part of result.

As for whether 20 μ M DFX has killing effect on cells, we conducted CCK-8 assay and apoptosis assay. We found that 50 μ M DFX had an inhibitory effect on cell proliferation, but did not induce an increase in apoptosis (Figure S2). Therefore, we think that 20 μ M DFX has an inhibitory effect on cell proliferation, leading to the reduction of nucleus.

We maintained the same concentration of the three iron chelators (DFX, DFO and DFP) in the initial experiment in order to initially reveal that the effect of anti-BKPyV may be related to iron chelation and to guide the subsequent dose-inhibition experiment and iron supplement experiment. To be sure, "Our data suggested that iron chelators exhibited different anti-BKPyV effects due to their ability to chelate iron..." is indeed imprecise and can only be drawn in combination with subsequent experimental results. We have made correction according to the reviewer's comments.

The authors use BKPyV and BKV interchangeably. Should be streamlined.

Reply: Thank you for your nice suggestion. We have changed all BKV to BKPyV in the manuscript.

Figure 2 uses LTag and VP1 in Western Blot and Figure 3 used VP1 and VP3 in Western Blot. Is there a specific reason to use those sets of structural and non-structural proteins?

Reply: Thank you for your rigorous consideration. In FIG. 2, we intended to investigate that weather iron chelating agent have the same inhibition effect on early and late protein, so we detected both structural and non-structural protein expression. In FIG. 3, we focused on verifying the production of BKV was related to the degree of iron elimination, and the expression of VP1 and VP3 structural proteins was used to detect virus production (*Antiviral Research* 178 (2020)).

Line 391: Captive cells = infected cells?

Reply: Thank you for your careful check. It is more appropriate to use "infected cells" here, which we have modified in the manuscript.

Line 401: "The next step should be to examine the anti-BKPyV effect of iron chelators in vivo." Use of clinically relevant concentrations and further studies on physiological effect of in vitro models first?

Reply: We totally understand your concern. And we have deleted this sentence from the manuscript.

Line 402: "Moreover, possible adverse reactions should also be considered since patients with BKPyV infection after kidney transplantation are often accompanied by anemia symptoms." This is the only sentence mentioning any possible clinical side-effects of using iron-chelators in clinical setting, without discussing current clinical applications (except that its currently used for hyperferremia).

Reply: Thank you for your nice suggestion. We have added a discussion of opportunities and challenges in the clinical application of iron depletion therapy according to the reviewer's comments.

Minor Comments:

Line 66: TAGs = Tags, to be consistent

Reply: Thank you for your nice suggestion. We have made correction according to the reviewer's comments.

Line 76 and 77: Add references

Reply: Thank you for your nice suggestion. We have added the reference in the appropriate place.

Line 93: Human bladder carcinoma cells (TCCSUP; HTB-5, ATCC)

Reply: Thank you for your careful check. We have made correction according to the reviewer's comments.

Line 242: the expression of early LTag protein...

Reply: Thank you for your careful check. We have made correction according to the reviewer's comments.

Line 357: depletion on BKPyV infection occurred

Reply: Thank you for your careful check. We have made correction according to the reviewer's comments.

Reviewer #2 (Comments for the Author):

The manuscript, Intracellular low iron exerts anti- BK polyomavirus effect by inhibiting

the protein synthesis of exogenous genes, by the Zhu lab is a fairly comprehensive, rigorous, and unique study of the role cellular iron plays during a virus infection. They tested three drugs in clinical use for their ability to chelate intracellular iron and correlated this with inhibition of virus gene expression and subsequent viral loads in tissue culture supernatant. The results were impressive with each drug inhibiting infection between 27 and 98%. This correlated with drugs ability to reduce intracellular iron. The effects could be reversed by providing iron exogenously.

Major concern:

They provide no data regarding the potential cytotoxic or cytostatic effects of these drugs on the cell. These controls would be critical for proper interpretation of the results.

Reply: Thank you for your nice suggestion and we totally understand your concern. We have added the data regarding the effects of these drugs on the cell apoptosis and proliferation (as shown in Figure S2).

Summary:

This is an important and unique study that could lead to promising treatments for polyomavirus associated diseases. The experiments are well-controlled, with the exception of controls for cytotoxic/cytostatic effects, and the conclusions are supported by the data.

October 14, 2021

Dr. Tongyu Zhu
Shanghai Public Health Clinical Center, Fudan University
Shanghai
China

Re: Spectrum01094-21R1 (Intracellular low iron exerts anti-BK polyomavirus effect by inhibiting the protein synthesis of exogenous genes)

Dear Dr. Tongyu Zhu:

Your manuscript has been accepted, and I am forwarding it to the ASM Journals Department for publication. Both reviewers agreed that the manuscript was much improved and they both accepted the paper. You will be notified when your proofs are ready to be viewed.

Sincerely,

Clinton Jones
Editor, Microbiology Spectrum

Journals Department
Supplemental figure: Accept